# Dose-Dependent Sorafenib-Induced Immunosuppression Is Associated with Aberrant NFAT Activation and Expression of PD-1 in T Cells

**DOI:** 10.3390/cancers11050681

**Published:** 2019-05-16

**Authors:** Renuka V. Iyer, Orla Maguire, Minhyung Kim, Leslie I. Curtin, Sandra Sexton, Daniel T. Fisher, Sarah A. Schihl, Gerald Fetterly, Stephan Menne, Hans Minderman

**Affiliations:** 1Department of Medicine, Roswell Park Comprehensive Cancer Center, Buffalo, NY 14263, USA; 2Flow and Image Cytometry Shared Resource, Roswell Park Comprehensive Cancer Center, Buffalo, NY 14263, USA; Orla.Maguire@RoswellPark.org (O.M.); Hans.Minderman@RoswellPark.org (H.M.); 3Department of Surgical Oncology, Roswell Park Comprehensive Cancer Center, Buffalo, NY 14263, USA; Minhyung.Kim@RoswellPark.org (M.K.); Daniel.Fisher@RoswellPark.org (D.T.F.); 4Laboratory Animal Shared Resources, Roswell Park Comprehensive Cancer Center, Buffalo, NY 14263, USA; Leslie.Curtin@RoswellPark.org (L.I.C.); Sandra.Sexton@RoswellPark.org (S.S.); 5Bioanalytics, Metabolomics, and Pharmacokinetics Shared Resource, Roswell Park Comprehensive Cancer Center, Buffalo, NY 14263, USA; Sarah.Schihl@RoswellPark.org (S.A.S.); Gerald.Fetterly@roswellpark.org (G.F.); 6Department of Microbiology & Immunology, Georgetown University, Washington, DC 20057, USA; Stephan.Menne@georgetown.edu

**Keywords:** sorafenib, hepatocellular carcinoma, NFAT1, immunosuppression

## Abstract

The multikinase inhibitor sorafenib is the only standard first-line therapy for hepatocellular carcinoma (HCC). Here, we report the dose-dependent effects of sorafenib on the immune response, which is related to nuclear factor of activated T cells 1 (NFAT1) activity. In vitro and in vivo experiments were performed with low and high doses of sorafenib using human T cells and spontaneous developed woodchuck HCC models. In vitro studies demonstrated that following exposure to a high dose of sorafenib the baseline activity of NFAT1 in T cells was significantly increased. In a parallel event, high dose sorafenib resulted in a significant decrease in T cell proliferation and increased the proportion of PD-1 expressing CD8+ T cells with NFAT1 activation. In the in vivo model, smaller tumors were detected in the low-dose sorafenib treated group compared to the placebo and high-dose treated groups. The low-dose sorafenib group showed a significant tumor growth delay with significantly more CD3+ cells in tumor. This study demonstrates that sorafenib has immunomodulatory effects in a dose- and time-dependent manner. Higher dose of sorafenib treatment was associated with immunosuppressive action. This observed effect of sorafenib should be taken into consideration in the selection of optimum starting dose for future trials.

## 1. Introduction

Hepatocellular carcinoma (HCC) is the fifth most prevalent type of cancer, and one of the leading causes of cancer-related death globally [1]. The overall incidence is rising, with over 700,000 new cases of HCC diagnosed each year worldwide [2]. In addition, the number of deaths from HCC per year is almost identical to the number of new cases, reflecting a high case fatality rate and emphasizing the pressing need for the development of better treatment modalities [3]. Despite advances in various treatment strategies, prognosis of patients with HCC remains poor with tumor recurrence rates of 50% at 3 years and 70% at 5 years with prevention remaining an area of unmet need [4,5].

Sorafenib (Nexavar^®^, Bayer, Pittsburgh, PA, USA) is currently the only systemic agent approved for use in HCC. It is an orally administered multikinase inhibitor. It inhibits multiple cell surface tyrosine kinases, vascular endothelial growth factor receptor (VEGFR)-1, VEGFR-2, VEGFR-3, platelet-derived growth factor-β, Fms-like tyrosine kinase-3, and downstream intracellular serine/threonine kinase in the mitogen-activated protein kinase (MAPK) cascade. These kinases are involved in tumor cell signaling, proliferation, angiogenesis, and apoptosis [6,7,8]. In addition to sorafenib preventing the growth and progression of micrometastases by inhibiting angiogenesis, it could be expected that sorafenib will also be potent enough to block residual or metastatic tumor cells independently of its antiangiogenic action. While the therapeutic effects of sorafenib in HCC are mainly thought to be associated with its pro-apoptotic and antiangiogenic properties, there is evidence that sorafenib also has immune modulatory effects albeit with seemingly conflicting results [9,10,11]. There are reports of sorafenib inducing higher proportions of regulatory T cells (Tregs) [12] as well as reports that it reduces Tregs numbers [9,10]. There is support for detrimental immunomodulatory effects of sorafenib, which includes a report of inhibition of the immunostimulatory capacity of dendritic cells [13]. Conversely, Chen and colleagues have reported that sorafenib relieves inhibition of effector T cells in the tumor microenvironment and augments antitumor activity, which supports beneficial immunomodulatory properties [9]. The seemingly contradictory data may originate from the use of different model systems, varying drug exposure conditions, or varying achieved drug concentrations between the periphery and the tumor microenvironment. 

A possible role of the immune system in the etiology of HCC is suggested by the increased presence of immunological factors favoring immunosuppression such as augmented numbers of Tregs, myeloid-derived suppressor cells (MDSC), PD-1+exhausted T cells, and increased levels of immunosuppressive cytokines in patients with HCC, compared with normal controls [14].

Nuclear factor of activated T cells (NFAT) is a family of transcription factors that includes 5 members, 4 of which are regulated by Ca^2+^ signaling; NFAT1 (NFATc2 or NFATp), NFAT2 (NFATc or NFATc1), NFAT3 (NFATc4), NFAT4 (NFATx or NFATc3), and calcineurin-independent NFAT5 (TonE-BP or NFATL1) [15,16]. Roles of NFAT in carcinogenesis with several malignancies were reviewed. The exact function of NFAT has been shown to have context-dependent functions in different malignancies [17]. The NFAT pathway has been reported to be involved in driving proliferation of HCC cells, with tumors exhibiting high expression of NFAT2 [18]. More recently, NFAT down-regulation in HCC has been shown to be associated with larger tumors, advanced metastasis and poor prognosis. The same study also reported that NFAT2 over-expression in HCC cell lines decreased proliferation and increased apoptosis by activating the Fas-ligand pathway [19].

The aim of this study was to investigate the dose-dependent effects of sorafenib on the immune response. Mechanistically, we focused on the NFAT transcription factor pathway as a well-established key regulator of T cell activation. The calcium-regulated NFAT transcription factors have key roles in the regulation of many aspects of T-cell function [16], and NFAT is known to contribute to hematopoiesis as well as innate and adaptive immune functions in a variety of myeloid lineage cells [20]. Especially, NFAT1 regulates the transcription of the polarizing cytokines IFN-γ and IL-4, which drive Th1/Th2 cell differentiation [21]. Also, TNF-α in T cells is directly regulated by NFAT1 through promoter occupation [22]. One of NFAT functions in immune cells is to regulate the expression of potent immunomodulatory cytokines. In T cells, the downstream targets of NFAT include IL-2 growth factor, which regulates lymphocyte development and activation and may contribute to the overall function of Tregs by regulating their expansion, differentiation, and survival [23]. NFAT is not only essential to T cell activation, but paradoxically is also involved with T cell “tolerance” and “anergy” [24,25]. In addition, NFAT participates in the transcriptional program of CD8+ T cell exhaustion [26]. The duality of immune-stimulatory versus -inhibitory action is dependent on the cell type and signaling context in which it is activated [27,28]. 

A model that spontaneously develops HCC is preferable to the implanted tumor models because it allows for the development of HCC that may exhibit immune escape properties. The woodchuck HCC model with woodchuck hepatitis virus (WHV) is a naturally occurring tumor model that is similar to human HCC caused by hepatitis B virus (HBV) [29]. In addition to in vitro experiments, the preclinical novel HCC woodchuck model was used to clarify roles of NFAT1 with sorafenib treatment. 

## 2. Results

### 2.1. Dose- and Time-Dependent Effects of Sorafenib on NFAT1 Nuclear Translocation in T Cells

The dose-dependent effects of sorafenib exposure on the nuclear translocation of NFAT1 were studied in healthy human T cells. Dosing was determined based on pharmacokinetic studies that indicated the plasma concentration of sorafenib in clinical trials reached approximately 2 mg/L. 

This corresponds to 4 µM. Low dose was set at 1 µM and high dose at 10 µM for in vitro experiments [30]. The data demonstrated that following exposure to high dose (10 µM) sorafenib in the absence of CD3/CD28 stimulation the baseline activity of NFAT1 in T cells was significantly increased. In the presence of CD3/CD28 stimulation, the high dose of sorafenib tested significantly inhibited the anti-CD3/CD28-induced NFAT1 activity (Figure 1A). 

This inhibitory effect of 10 µM sorafenib on exogenously-induced T cell stimulation was also observed when phorbol 12-myristate 13-acetate (PMA)/ionomycin was used to stimulate the cells but to a lesser extent (Appendix A), suggesting that the sorafenib inhibitory action in activated T cells is TCR-associated. In non-stimulated CD8+ T cells, the degree of NFAT1 activation by high dose (10 µM) sorafenib was time dependent. Within the time frame studied, the most significant increase in nuclear translocation of NFAT1 was observed at 4 h following exposure to 10 µM sorafenib while no significant increase was observed following 1 h exposure (Figure 1B). Similar results were detected in CD4+ T cells (Figure 1C,D).

### 2.2. Dose-Dependent Effects of Sorafenib on T Cell Proliferation

In order to determine if the sorafenib-induced effects on NFAT1 activity had a functional consequence, mixed lymphocyte reactions (MLRs) were performed in the presence of sorafenib. Human cells from two healthy donors were mixed in a one-way MLR with increasing concentrations of sorafenib. The data demonstrated that a significant decrease in the MLR response was observed only in the presence of the high dose of sorafenib (Figure 1E). Thus the sorafenib-induced base-line activation of NFAT1 and corresponding inhibition of CD3/CD28-induced activation of NFAT1 (Figure 1A) correlates with a reduced proliferative functional response. In addition, the identical experimental was performed with woodchuck cells. The data showed in accordance with the human MLR results. A sorafenib-induced inhibition of the woodchuck MLR response was observed with the notable difference that a dose responsiveness of the inhibition was more evident (Figure 1F). MLRs were chosen to confirm the in vitro results observed in healthy donor assays due to the limited availability of woodchuck specific antibodies. Human antibodies were tested to determine whether they cross-reacted with woodchuck peripheral blood but results were negative.

### 2.3. Effects of High Dose Sorafenib on PD-1 Expression on T Cells

High dose sorafenib treatment increased the proportion of PD-1-expressing CD8+ and CD4+ T cells in blood without exogenous stimulatory factors (Figure 2A). In these PD-1+ cells, NFAT1 activation was increased (Figure 2B) and CD8 expression decreased (Figure 2C). In contrast, this NFAT1 activation and decreased expression of CD8 was not observed in PD-1 negative T cells that had been exposed to the same high dose of sorafenib (Appendix A). A decrease in the proportion of CD4+CD25^bright^ T cells, a phenotype consistent with regulatory T cells, was detected with sorafenib treatment in a dose-dependent manner (Figure 2D). 

### 2.4. Pharmacokinetics with Different Doses of Sorafenib Treatment

Clinical studies of patients that undergo multiple dosing of oral sorafenib have been associated with plasma concentration-time profiles that have high interpatient pharmacokinetic variability and are not proportional with increasing dose [31].

The plasma concentration-time profiles of sorafenib in woodchucks were not dose proportional over the course of multiple dosing with oral sorafenib (Figure 3A), which is a result consistent with clinical studies. After dosing on Day 1, the sorafenib concentrations in plasma were significantly higher in the high dose group versus the low dose group. This dose-dependent effect is significantly decreased by Day 5 and decreased further through Days 12 and 19 until the measured sorafenib concentration between the two dose groups are the same within error. This result is confirmed by calculation of the maximum plasma concentration normalized by dose (C_max_/D) for each dose group on Day 1 and Day 19 of the pharmacokinetic study (Table 1). The C_max_/D of the high dose group is ~8 times greater than the low dose group on Day 1, but by Day 19 the C_max_/D for both dose groups is the same. This confirms that as the multiple dose study continued through Day 19, increasing the dose from 2.5 to 5.0 mg/kg did not result in higher plasma concentrations of sorafenib. But, the resulting pharmacokinetic parameters (area under the curve (AUC), C_max_, T_max_) all had higher variability (%CV) for the high dose group than was observed in the low dose group (Table 1).

### 2.5. Toxicity of Sorafenib Treatment In Vivo Model of HCC

Any distinct general performance change of animals was not detected such as moving activity, appetite, defecation, urination, and weight changes, and no dose interruptions of treatments occurred during the course of experiments in all treatment groups. In addition, no statistical difference was detected in CBC and biochemistry analyses among the three groups in the end of experiment (Appendix A). 

### 2.6. Efficacy of Sorafenib Against HCC

First, we compared the preventive effects of long-term treatment with low (2.5 mg/kg/day) and high (5.0 mg/kg/day) doses of sorafenib against HCC development. 100% of animals (15 woodchucks) developed HCC regardless of doses of sorafenib, and median time to tumor development was 295 days, 384 and 351 days in the placebo, low and high dose sorafenib groups, respectively (*p* = 0.35) (Figure 3B). Even though there was no statistical difference in median time of tumor burden among groups, when tumor was detected first by ultrasound, tumor size was smaller in the low dose group compared to the placebo and high dose groups (Figure 3C). Second, we investigated whether sorafenib has any effect on delay of tumor growth after HCC development. From the first detection of tumor to being 2 cm HCC, median time was 55 days for placebo, 111 days for 2.5 mg/kg and 42 days for 5 mg/kg groups. The low dose sorafenib group showed a significant tumor growth delay compared to the placebo and high dose sorafenib groups (*p* = 0.024), and the high dose sorafenib group failed to show any tumor growth delay effect compared to the placebo group (*p* = 0.5) (Figure 3D). Third, the necrotic range of HCC was evaluated with H&E stain after tumor size reached over 2 cm. There was a significantly large necrotic change in the high dose sorafenib group compared to the placebo (*p* = 0.02), but there was no statistical difference between the high and low dose sorafenib groups (*p* = 0.07) (Figure 3E). In addition, the efficacy of sorafenib with a short-term usage was evaluated. Woodchucks in the low and high dose groups (*n* = 3 animals/group) were treated with sorafenib for 90 days. The finding of tumor growth delay with a low dose of sorafenib compared to the high dose group was not achieved after the short-term sorafenib usage (Figure 3F), suggesting that long-term low dose treatment may be necessary to see a positive impact. 

One of the few commercial antibodies that cross-reacts with woodchuck antigens is a mouse anti-rat CD3. HCC was stained with CD3 antibody in the end of in vivo experiments. More CD3+ cells were infiltrated into tumors with the low dose sorafenib treatment compared to the high dose. However, there was no statistical difference in the CD3+ cell infiltration in tumors between the placebo and low dose sorafenib groups (Figure 4).

## 3. Discussion

Although sorafenib is the first molecular-targeted agent that was shown to increase the survival rates of HCC patients [32], the mechanism underlying sorafenib-mediated antitumor activity has not been fully characterized. When examined as a secondary chemopreventive agent for cancer, it was unsuccessful, and sorafenib was treated at full dose in the sorafenib as adjuvant treatment in the prevention of recurrence of hepatocellular carcinoma (STORM) trial [33]. There is increasing evidence that sorafenib is not only an antiangiogenic tyrosine kinase inhibitor, but that it also affects immune function [10,34]. In this study, we investigated the immune modulation effects of low and high doses of sorafenib in the absence of exogenous stimulatory factors such as CD3/CD28 or PMA/Ionomycin. The immune regulator NFAT1 was activated with a high dose sorafenib, which is clinically applicable, but this NFAT1 activation was not observed with lower doses of sorafenib. NFAT1 activation in the absence of co-stimulatory factors has been associated with T cell anergy [35]. In agreement with this paradigm, the high dose sorafenib exposure resulted in significant inhibition of CD3/CD28-induced NFAT1 activation, and inhibition of a proliferative T cell response in a MLR. 

The critical role of the PD-1 pathway as a deterrent to anticancer immunity has now been validated in clinical studies using anti-PD-1 monoclonal antibody, and this will likely be a paradigm shift in HCC [36]. Engagement of PD-1 on T cells with PD-L1 on tumor cells downregulates anticancer T cell responses [37]. Upregulation of PD-L1 by neoplastic cells allows cancers to escape the anticancer effects of T cell responses [38]. Recently published studies in a murine HCC model [39] and in humans [40] showed a reduction in PD-1-expressing CD8+ T cells with sorafenib treatment. In our study the NFAT1 activation by high dose sorafenib was associated with an increase in the proportion of PD-1+ T cells and a reduction of CD8 expression in the absence of exogenous stimulatory factors. Thus, our observations can be harmonized with the previous literature (showing the reduction of the number of CD8+ T cell following sorafenib exposure) [39,40] regardless of PD-1 expression. The added information that our experiments contribute is that the number of CD8+ T cell reduction is due to down regulation of CD8 expression, and this occurs concomitantly with upregulation of PD-1 expression and activation of NFAT1 when cells are not with exogenous stimulatory factors. In this context, it is important to note that the NFAT pathway has been implicated in the regulation of tolerance in CD8 but not CD4 cells [25]. 

In the current study T cells were phenotyped for CD4 and CD25 co-expression; a phenotype of CD4 positive, CD25 bright is consistent with functional immunosuppression attributed to T regulatory cells [41]. The observed sorafenib-induced reduction of CD4+CD25^bright^ T cells is in agreement with previously published literature [10] but is contradictory to others [42,43]. Definitive identification of this population as T regulatory cells would require additional functional markers, such as FoxP3 [44]. 

To determine if a dose reduction could be a better tolerated and allow evaluation of chemopreventive effects of sorafenib on HCC, an established woodchuck model of spontaneous HCC development was used as it has a very long latent period of HCC development. The eastern woodchuck (*Marmota monax*) has been demonstrated to be a useful animal model for the study of HBV associated HCC [29]. Chronic hepatitis can be induced in these animals by infection with WHV, a virus that shares a genomic structure and biological properties similar to human HBV [45]. WHV infected woodchucks were supplied by two different institutes with the same strain and virus. The woodchuck HCC usually developed within 2 years after WHV infection. The tumor sizes of woodchucks were more suitable for imaging diagnosis than those of smaller rodents such as mice and rats. Two different diluents were used to dissolve sorafenib in this translational study, and we confirmed drug concentrations and its activity before the study. Although patients have been treated with sorafenib twice a day in most clinical trials, woodchucks were treated once a day in this study, because, based on pharmacokinetic studies, the drug concentrations in woodchucks were stable due to a long half-life of sorafenib. Even though a big difference of sorafenib concentrations between low and high doses was measured in peripheral blood 1 day after treatment, low and high doses of sorafenib reached a similar drug concentration at steady state. This finding was also described in the four phase I trials. Multiple dosing with oral sorafenib was associated with high interpatient pharmacokinetic variability, and plasma concentration-time profiles of sorafenib increased less than proportionally with increasing dose [31,46]. Steady-state concentrations of sorafenib were reached after 7 days treatment, with no further substantial accumulation observed after this time [47]. Although there appeared to be a greater likelihood of anticancer activity with over 200 mg bid, there was no clear dose-dependent relationship on anticancer activity in the range of 200 mg to 600 mg bid [48]. However, we anticipate that there would be different drug concentrations in tumor based on treated doses, and higher dose sorafenib will have immune suppression effects in the tumor.

It is known in humans that anti-HBV therapy can prevent HCC [49], and this preclinical study shows that sorafenib treatments did not prevent HCC or inhibit WHV replication compared to the placebo group. However, there was a significant difference on the initial tumor volumes and tumor growth speeds after HCC development between the placebo and the group treated with the low dose (2.5 mg/kg) of sorafenib. This tumor growth delay and lower initial tumor volume using low dose sorafenib was not detected with a short-term usage of the drug, making the case for long term therapy for this effect to occur rather than a defined period of therapy. Interestingly, a high dose of sorafenib did not achieve any statistical difference on initial tumor volume and tumor growth speed compared to the placebo group. Taking this together with the in vitro data, these observations are consistent with a possible dose-dependent immune modulatory role of sorafenib that is beneficial at low dosages but detrimental at high dosages.

The STORM trial concluded that sorafenib was not an effective intervention in the adjuvant setting for HCC following resection or ablation [33]. However, these results should be interpreted with caution, since treatment interruptions and dose modification occurred due to a relatively high number of drug-related adverse events. This raised the question of the optimum starting dose of sorafenib in patients and median duration of treatments was 1 year. 

Even though significant tumor necrotic changes were detected in the high dose sorafenib group compared to the placebo group, there was no benefit for tumor growth. Sorafenib is multikinase inhibitor including VEGFR, so sorafenib treatment could induce necrotic changes in tumors with the drug effect itself. However, tumor hypoxia is closely correlated with tumor size. As tumors enlarge, the fraction of hypoxic cells is increased [50], and measured values of tumor partial pressure of oxygen (pO_2_) are decreased [51]. It is also well established that tumor necrosis increases with increasing tumor volume [52,53]. So, it is plausible that this necrotic change was not the drug effect but the necrotic formation by depletion of substrates or accumulation of metabolic waste products. 

The cellular pharmacokinetics of sorafenib were not described in this study; however, more infiltrating T cells were detected with low dose in vivo sorafenib treatment compared to high dose sorafenib in tumor, consistent with differential immunomodulatory effect between the two doses used. 

## 4. Materials and Methods

### 4.1. Chemicals

In vitro dosing: 0.1, 1, 5, 10 µM of sorafenib (Bayer, Leverkusen, Germany) were prepared from 1 mM stock solution stored in DMSO at −80 °C. Final concentrations of drug were prepared in AIM-V serum free medium (Thermo Fisher Scientific, Grand Island, NY, USA) and repeated freeze-thaws avoided.

### 4.2. Isolation of Peripheral Blood Mononuclear Cells

Heparinized peripheral blood was collected by venous puncture from human healthy donors and healthy woodchucks. Peripheral blood was collected in accordance with protocols approved by the Institutional Review Board (IRB) (I 36404, 7 September 2017 of approval) and Institute Animal Care and Use Committee (IACUC) (1064W, 11 July 2015 of approval) at Roswell Park Comprehensive Cancer Center. Peripheral blood mononuclear cells (PBMCs) were isolated by density gradient centrifugation using Ficoll-Hypaque (MilliporeSigma, St. Louis, MO, USA).

### 4.3. Analysis of NFAT1 Nuclear Translocation Activity

PBMCs were allowed to ‘recover’ in AIM-V medium (Thermo Fisher Scientific) at 37 °C, 5% CO_2_ in a humidified atmosphere overnight. 2 × 10^6^ cells were treated with 0–10 µM sorafenib for the durations specified in the results section. If the experimental conditions required T cell stimulation, samples were co-treated (stimulated) with 1 µg/mL anti-CD3 and 1 µg/mL anti-CD28 for 2 h. Following treatment, PBMCs were placed on ice to stop drug action and immunophenotyped for CD4, CD8, CD25, and PD-1 (CD279). NFAT1 was detected by indirect labeling. Antibodies were diluted in permeabilization wash buffer (PWB) consisting of 0.1% Triton-X-100 in FCM Buffer (1× phosphate buffered saline (PBS) + 0.5%BSA). The primary Rabbit polyclonal NFAT1 antibody was diluted 1:50 in PWB. Following this, 1:200 dilution of secondary AF647 conjugated F(ab’)2 fragment donkey anti rabbit IgG antibody (Jackson ImmunoResearch Laboratories Inc., West Grove, PA, USA) was added. Just prior to data acquisition on the ImageStream, DAPI was added to all samples (0.5 µg/mL final concentration), to stain the nucleus.

### 4.4. Imaging Flow Cytometry

Imaging flow cytometry acquisition, compensation, and analysis was performed on an imaging flow cytometer (ImagestreamX Mk-II; Amnis, part of MilliporeSigma, Seattle, WA, USA) as described previously [54,55]. Following compensation for spectral overlap, image analysis was performed with IDEAS^®^ software (Amnis, Seattle, WA, USA). To assess nuclear NFAT1 translocation, the corresponding nuclear image and NFAT1 image of each cell was compared and a Similarity Score (SS) was assigned for individual cells using published methods (Appendix A) [54,55]. 

### 4.5. Flow Cytometry

Flow cytometry acquisition and compensation was performed on a LSR Fortessa (BD Biosciences, San Jose, CA, USA). Analysis of listmode data was performed using Winlist 8.0 (Verity Software House, Inc., Topsham, ME, USA). The gating strategy is shown in Appendix A. Briefly, the viable responder lymphocytes were hierarchically gated for single, low side scatter, viable cells with absent CD45 labeling. The percentage of proliferating cells was determined based on CFSE dye dilution compared to the CFSE labeling intensity from the non-proliferating MLR negative control sample. The full list of antibodies and reagents used in assays is available in Appendix A.

### 4.6. T Cell Responses in Mixed Lymphocyte Reactions (MLRs)

Mixed lymphocyte reactions were performed in human (human vs. human, Figure 1E) and woodchucks (woodchuck vs. human, Figure 1F). 10 mL peripheral blood from healthy donors and 3 mL woodchuck peripheral blood isolated. Stimulator cells (human cells for both assays) were irradiated at 40 Gy in a Mark I Cesium-137 irradiator for 20 min. Following irradiation, cells were stained for CD45-APC (BD Biosciences, San Jose, CA, USA). Responder cells (non-irradiated human or woodchuck cells) were labeled with CFSE (Thermo Fisher Scientific, Eugene, OR, USA). Target and responder cells were placed in a round bottom 96 well polypropylene plate at a 2:1 ratio in the presence of 0–10 µM sorafenib and incubated at 37 °C for 96 h. Following the incubation, cells were stained with LiveDead Violet (Thermo Fisher Scientific) to facilitate dead cell exclusion.

### 4.7. Dosing Solution Preparation

#### 4.7.1. Long-Term Efficacy Study

Sorafenib tosylate (Bayer Pharmaceuticals, Berlin, Germany) was used for the preparation of dosing formulations in this study. Formulations were prepared at a concentration of 5 mg/mL, adjusting for the salt form. The sorafenib was weighed and slowly mixed into ethanol and then sonicated for 5 min to fully disperse the powder. Methylcellulose (0.5%) (Fisher Scientific, Hampton, NH, USA) or carboxymethylcellulose (0.5%) (Millipore Sigma, Burlington, MA, USA) was slowly added to the solution with constant mixing to the required final volume.

#### 4.7.2. Short-Term Efficacy Study

Sorafenib *p*-toluenesulfonate salt (LC Laboratories, Woburn, MA, USA) was used with carboxymethylcellulose (0.5%) for the preparation of dosing formulations in this study.

### 4.8. LC-MS/MS and Noncompartmental Analyses

Woodchuck plasma study samples were analyzed for sorafenib with an LC-MS/MS assay using authentic standards of sorafenib and the internal standard (IS), d_3_-sorafenib, both obtained from Toronto Research Chemicals (Toronto, ON, Canada). Study samples were quantitated using calibration and quality control (QC) samples prepared by spiking sorafenib into sodium heparinized human plasma (Bioreclamation, LLC, Westbury, NY, USA). The woodchuck plasma study samples were extracted using a Quadra 4 SPE Automated Liquid Handling System (Tomtec Inc., Hamden, CT, USA) by mixing a 50 µL aliquot of a calibrator, QC, plasma blank, or study sample with 500 µL of ice cold acetonitrile containing the IS for the LC-MS/MS. LC-MS/MS analysis was performed using a Finnigan Surveyor MS Pump Plus and Thermo Pal Autosampler System interfaced with a TSQ Quantum Ultra Mass Spectrometer, all obtained from Thermo Fisher Scientific (Waltham, MA, USA). A noncompartmental pharmacokinetic analysis (NCA) was performed on individual concentration-time data using a linear trapezoidal, linear/log interpolation calculation method with uniform weighting. The software program utilized for this analysis was Phoenix 64 WinNonlin Version 6.3 (Pharsight Corp., St. Louis, MO, USA). Study sample (*n* = 9) concentrations that were above the lower limit of quantitation (LLOQ) of the assay (62.5 ng/mL) following dose administration (Day 1: pre-dose and 1, 3, 5, 7 and 24 h; Day 19: pre-dose and 1, 3, 5, and 7 h) were used in the NCA. 

### 4.9. Animal Tumor Model and Screening of Tumors

Male and female eastern woodchucks *(Marmota monax)* from two different institutes were used. WHV-infected woodchucks from Cornell University (15) were used for a long-term efficacy study, and six woodchucks in a short-term efficacy study were bred at Roswell Park Comprehensive Cancer Institute Center. Woodchucks were infected at 3–5 days of age with dilute serum from standard infectious pools derived from WHV carriers. Pups were inoculated with diluted, pooled serum from WHV^+^ animals. The inoculum was prepared from pooled serum of WHV^+^ animals that is diluted aseptically 1:10 in PBS. In a biological safety cabinet, each pup is manually restrained and 100 microliters of inoculum is injected subcutaneously in the intrascapular region. Chronic infection was verified at approximately 3, 6, and 9 months of age by WHV viral titer status. Once the pups are confirmed WHV positive, they were monitored by ultrasound (Philips HDI 5000, Amsterdam, The Netherlands) to confirm the presence of HCC. 

### 4.10. In Vivo Study Procedures

In total, 21 animals bearing one or more liver tumors were included in this study for 2 separate experiments. While receiving sorafenib or placebo, animals were monitored with ultrasound every 2 weeks for tumor development greater than 5 mm in diameter. During follow-up by ultrasound, the same section plane was examined to ensure comparable measurements of the tumor size. Tumor volume was calculated using this formula: volume = shortest axis^2^ × longest axis × 0.52. 

### 4.11. In Vivo Animal Dosing with Sorafenib

All WHV carrier woodchucks were randomly divided into three groups such as placebo, low dose (2.5 mg/kg/day), and high dose (5.0 mg/kg/day) groups (*n* = 5 animals/group) in a long-term efficacy study based on clinical and preclinical studies [56,57]. Also, a compartmental pharmacokinetic (PK) model was used to select the two lowest doses that can achieve exposure sufficient to inhibit pharmacodynamic targets. Experimental woodchucks were treated with placebo, 2.5 mg/kg, or 5.0 mg/kg sorafenib once daily with a 5 days on and 2 days off schedule. Animals were dosed orally until tumor development. Another experiment was performed with the low (2.5 mg/kg/day) and high (5.0 mg/kg/day) dose groups (*n* = 3 animals/group) to see the efficacy of short-term (90 days treatment) usage of sorafenib. The primary endpoint of the in vivo study was disease free survival (DFS), and DFS was measured from the date of randomization until the date of any liver tumor detection. The secondary endpoint was tumor size over 2 cm, which was measured from the date of first tumor detection. All experimental protocols were approved by IACUC.

### 4.12. Assessment of Treatment Toxicity

General performance statuses of animals were evaluated such as moving activity, appetite, defecation, urination, and weight changes regularly. In addition, CBC and blood biochemistry were measured two times per month during the course of experiment by ProCyte Hematology and Catalyst Chemistry Analyzers (IDEXX, Hoofddorp, The Netherlands). 

### 4.13. Assessment of Tumor Responses to Treatment

#### Immunohistochemistry (IHC) and Histological Staining

Tumor specimens were diagnosed as HCC based on the established classification of neoplastic liver lesions. After standard processing and embedding in paraffin, 4 μm sections were prepared, deparaffinized, stained with hematoxylin-eosin (H&E), studied with a Zeiss Axio Imager A1 microscope (Carl Zeiss, Oberkochen, Germany) by a pathologist blinded to the treatment arm. Morphometric study of areas of necrosis was compared to the size of the tumor sections by counting view fields at medium objective magnification. IHC staining with mouse anti-rat CD3 (BD Biosciences, San Jose, CA, USA) was performed on 5 µM paraffin sections of posttreatment tumor tissues obtained after tumor reached over 2 cm. Images of at least 7 randomly selected fields (unit area of each field, 0.34 mm^2^) were captured by observers blinded to sample identity. Identical exposure times and image settings were used within each experiment. Images were analyzed with ImageJ software (http://rsb.info.nih.gov/ij, Bethesda, MD, USA).

### 4.14. Statistical Analyses

All data are shown as mean ± standard deviation of mean. Flow cytometry results, tumor volumes, tumor volume changes, WHV titers and necrotic changes were assessed by 2-tailed paired Student’s *t*-test. Statistical analyses of DFS and tumor growth delay were performed with GraphPad Prism, version 7 software (GraphPad Software, Inc., La Jolla, CA, USA). Values of *p* < 0.05 were considered significant. 

## 5. Conclusions

This study demonstrated that sorafenib has immunomodulatory effects with dose- and time-dependent manner. To our knowledge, the results are the first to demonstrate that a high dose of sorafenib induces NFAT1 activation, which is accompanied by an immunosuppressive phenotype including inhibition of T cell proliferation and increase in PD-1 expression. It did not occur with low dose sorafenib. In addition, higher dose sorafenib treatment was associated with a higher tumor growth rate with an immunosuppressive action in a translational in vivo HCC model. Furthermore, these studies also showed the tolerability, a slower tumor growth rate, and a smaller initial tumor size after long-term treatment with low dose sorafenib, where the immunosuppressive effects were not seen. Placing this work into human context, the effect of anti HBV therapy on delaying HCC seen in woodchucks has been translated to similar benefit in delay of secondary HCC recurrence in humans [29]. Our work suggests that the negative results of the large phase 3 adjuvant STORM study [33] done in humans where effect of full dose sorafenib compared to placebo in delaying HCC recurrence following surgery or ablation may have been due to poor tolerability as well as possibly the dose selected. The observed immunosuppressive effects of sorafenib should be taken into consideration in the selection of optimum starting dose for future trials. 

## Figures and Tables

**Figure 1 cancers-11-00681-f001:**
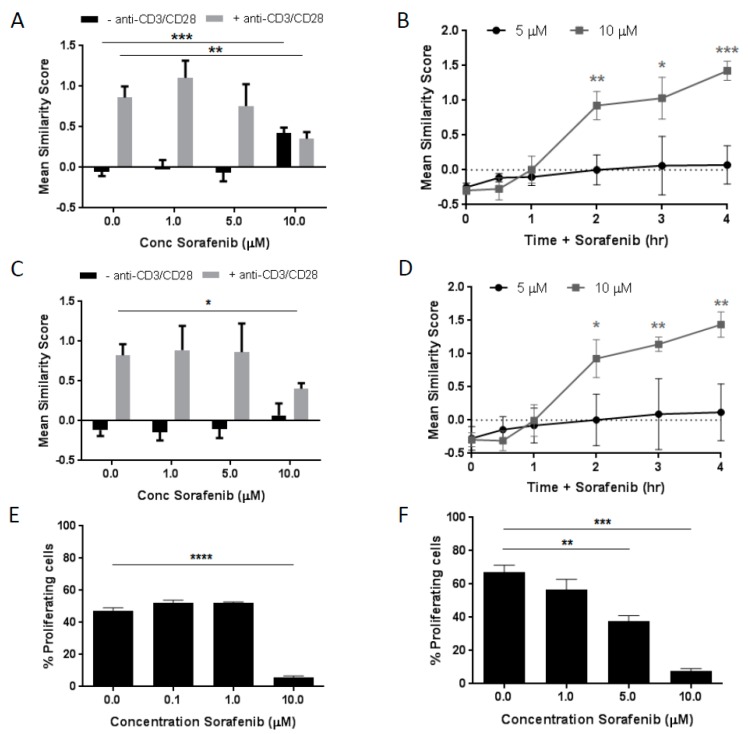
Sorafenib has dose- and time-dependent effects on nuclear factor of activated T cells (NFAT) nuclear localization in T cells. (**A**) NFAT1 nuclear localization, measured by mean similarity score by imaging flow cytometry, exhibits a dose-dependent increase without T-cell receptor (TCR) stimulation (-anti-CD3/CD28 antibodies), and decreases with anti-CD3/CD28 antibodies in CD8+ T cells, (**C**) similar results observed in CD4+ T cells. (**B**)Mean similarity score has a time-dependent increase with high dose sorafenib (10 µM) in CD8+ T cells, (**D**) similar results observed in CD4+ T cells. (**E**) Graph shows results from one-way mixed lymphocyte reaction (MLR) in human T cells stained with carboxyfluorescein succinimidyl ester (CFSE) in the presence of increasing sorafenib concentrations. (**F**) Graph shows results from a similar one-way mixed lymphocyte reaction (MLR) in woodchuck T cells. All experiments are at least *n* = 3, * *p* < 0.05, ** *p* < 0.01, *** *p* < 0.001 by Student’s *t*-test.

**Figure 2 cancers-11-00681-f002:**
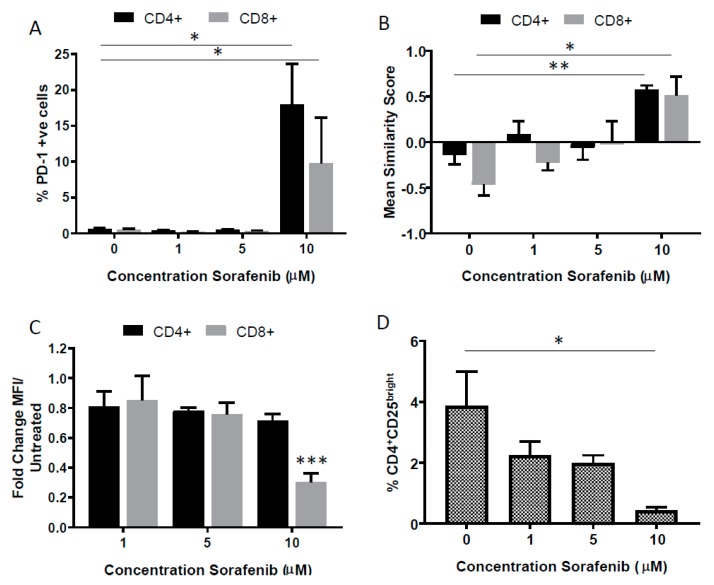
High dose sorafenib affects PD-1 expression on T cells. (**A**) % PD-1+ CD4+ cells (black bars) and % PD-1+CD8+ cells (grey bars) increases with high dose sorafenib (10 µM), measured by imaging flow cytometry, (**B**) These PD-1+ cells exhibit an increased NFAT1 nuclear localization (measured by mean similarity score), (**C**) These PD-1+ cells exhibit an decreased CD8+ protein expression with high dose sorafenib (10 µM), (**D**) the % CD4+CD25^bright^ cells, as a % of total CD4+ cells decreases. Experiments are *n* = 3, * *p* < 0.05, ** *p* < 0.01, *** *p* < 0.001 by Student’s *t*-test.

**Figure 3 cancers-11-00681-f003:**
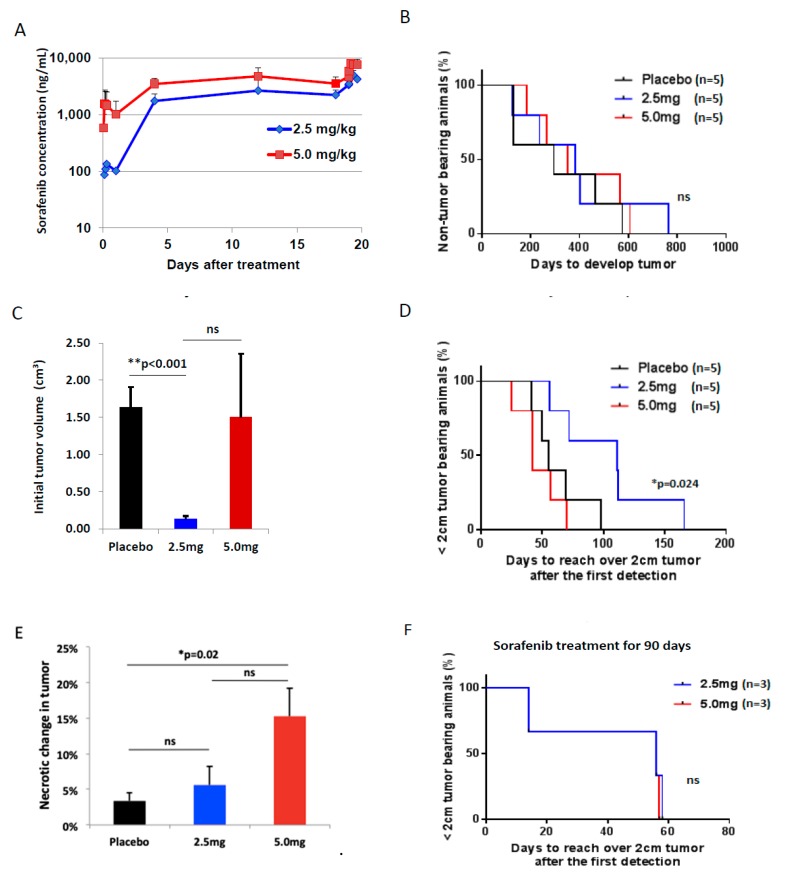
In vivo study with woodchuck hepatitis virus (WHV) infected woodchucks. (**A**) Graph plot of mean sorafenib concentration-time data for the low (2.5 mg/kg) and high (5 mg/kg) dose groups with a 5-days-on and 2-days-off schedule. (**B**) percent of non-tumor bearing animals after sorafenib treatment with different doses of sorafenib (*n* = 5), ns; not significant. (**C**) initial tumor volumes when tumor was detected for the first time, ** *p* < 0.001 by Student’s *t*-test, ns; not significant. (**D**) percent of less than 2 cm sized tumor bearing animals after the first tumor detection (*n* = 5), * *p* < 0.05 by Log-rank (Mantel-Cox) test. **E,** percent of necrotic change in tumor after tumor reached over 2 cm. (**F**) percent of less than 2 cm sized tumor bearing animals after a short-term (90 days) usage of 2.5mg/kg or 5.0 mg/kg sorafenib (*n* = 3), ns; not significant.

**Figure 4 cancers-11-00681-f004:**
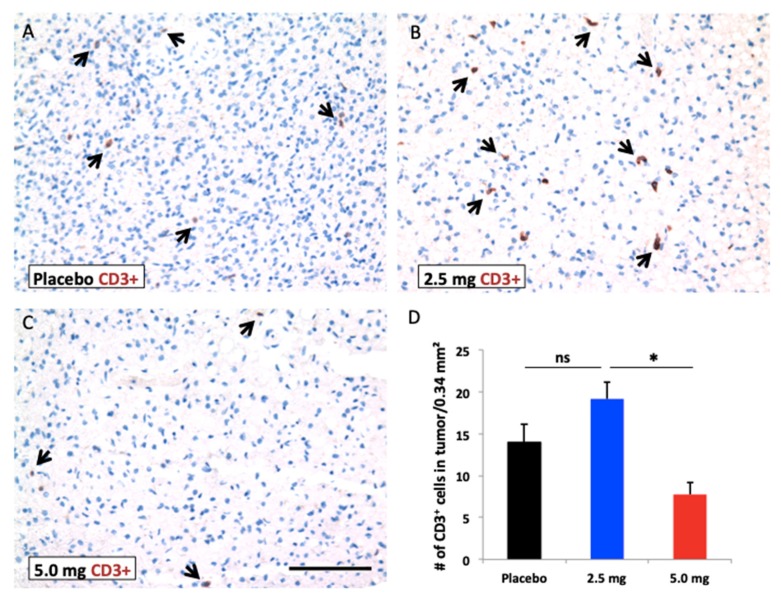
Immunohistochemistry with CD3 antibody. (**A**–**C**) representative photos after different doses of sorafenib treatment in the end of study (Arrow; CD3^+^ T cell). Scale bar: 100 μm. (**D**) numbers of CD3^+^ T cells in tumors (*n* = 3), * *p* < 0.05 by Student’s *t*-test, ns; not significant.

**Table 1 cancers-11-00681-t001:** Calculated noncompartmental pharmacokinetic parameters for Day 1 (pre-dose, 1, 3, 5, 7, and 24 h) and Day 19 (pre-dose, 1, 3, 5, and 7 h).

PK Parameters	AUC_All_ (h * ng/mL)		AUC_1–7 h_ (h * ng/mL)		C_max_ (ng/mL)		C_max_/D (ng/mL)/(mg/kg)		T_max_ (h)	
	*n*, Geo Mean (%CV)		*n*, Geo Mean (%CV)		*n*, Geo Mean (%CV)		*n*, Geo Mean (%CV)		*n*, Geo Mean (%CV)	
Dose Level (mg/kg)	Day 1	Day 19	Day 1	Day 19	Day 1	Day 19	Day 1	Day 19	Day 1	Day 19
2.5	2, 2600 (4.76)	3, 30,300 (15.8)	2, 588 (0.648)	3, 26,800 (15.8)	2, 136 (3.41)	3, 4970 (19.8)	2, 54.4 (3.41)	3, 1990 (19.8)	2, 5.92 (23.6)	3, 4.22 (26.6)
5.0	3, 21,000 (70.9)	3, 43,8000 (67.0)	3, 5960 (72.6)	3, 39,100 (65.5)	3, 1170 (73.7)	3, 7380 (64.5)	3, 433 (75.8)	3, 2000 (54.5)	3, 3.98 (53.3)	3, 3.98 (53.3)

AUC_All_: area under the curve including all time points, CV: coefficient of variance, *: hour, C_max_: maximum plasma concentration, C_max_/D: maximum plasma concentration/dose level, PK: pharmacokinetic, T_max_: time of maximum plasma concentration.

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
