# Peer review of "Dose-Dependent Sorafenib-Induced Immunosuppression Is Associated with Aberrant NFAT Activation and Expression of PD-1 in T Cells"

_cancers, 2019, doi:10.3390/cancers11050681_

Round 1
Reviewer 1 Report
In the manuscript entitled “Dose-dependent sorafenib-induced 3 immunosuppression is associated with aberrant 4 NFAT activation and expression of PD-1 in T cells”, the authors describe the role of high dose sorafenib in inducing immunosuppression in T-cells. The findings are described clearly and the data does suggest effects of high dose sorafenib in inducing immunosuppression via NFAT1 pathway.
There are no major problems found in the paper. However, it would have been preferable if the authors described better how they determined that sorafenib high dose is 10uM and correspondingly in vivo dose is 5mg/kg. If this is based on previous studies, that should be referenced. Are doses higher than the above limit tested? Do they exhibit toxicity? That should be discussed too.
Minor point:
The immunofluorescence data for CD3 in figure 3 does not look very convincing, the picture quality not very good atleast in this PDF.
Author Response
The authors wish to thank the reviewer for the favorable review and agree that the choice of dose need to be explained more clearly. We have now included this information on line 140 for in vitro experiments and line 515 for in vivo experiments with additional references.
We agree that the CD3 fluorescence images in figure 4 could be of better quality. We performed immunohistochemistry (IHC) and reanalyzed data in figure 4.
Reviewer 2 Report
In this paper, in vitro and in vivo experiments were performed to test the effects of low snd high doses of sorafenib on human T cells and spontaneous developed woodchuck HCCs. In vitro exposure to the high sorafenib dose induced a rise of the baseline activity of NFAT1 in T cells. A significant decrease in T cell proliferation and increase of the proportion of the PFD1 expressing CD8+ T cells with NFAT1 activation occurred in the presence of the high sorafenib dose. In the in vivo model, smaller tumors were only observed in the low-dose sorafenib group compared to placebo and high-dose groups, and a tumor growth delay occurred compared to other groups with significantly more CD3+ cells in the tumor. It is concluded that the immunomodulatory effect of sorafenib varies in a dose- and time-dependent manner and that the sorafenib treatment has an immunosoppressive effect.
This research has been well done and the results are appropriately described. Unfortunately, it does not really add new information on jhe mechanisms of the sorafenib therapeutic effect.
Author Response
We understand your concern. But, this study shows the value of sorafenib dose in chemoprevention and the differential effects on tumor growth rate in a very relevant animal model that may allow readers to design future studies. In addition, there are many new tyrosine kinase inhibitors now approved and the impact on early disease may be a clinical question that this unique animal model could help answer.
Reviewer 3 Report
Dear Authors,
Congratulations of well done work.
My suggestions:
Introduction: Add informations about the role of NFAT in liver carcinogenesis, it's expression in HCC tissues. Please, specify the aim of the study precisely.
Results: Would you explain why MLR was performed in blood from only two healthy donors (Materials and methods) and Figure 1 Graph E (Results) shows significant difference - The last sentence inform that all experiments are at least n=3.
Discussion: Please add your own opinion about results and the possibilities of using research results in clinical practice.
Conclusions:Consider rewording the conclusions to be clearly useful in further research on HCC pathogenesis or therapy, please.
Author Response
Introduction: Add informations about the role of NFAT in liver carcinogenesis, it's expression in HCC tissues. Please, specify the aim of the study precisely.
The role of NFAT in carcinogenesis has now been included in the paragraph beginning on line 79. Also, we have edited the next paragraph to directly outline the aim of the study in investigating the role of NFAT in the immune response to sorafenib (line 89).
Results: Would you explain why MLR was performed in blood from only two healthy donors (Materials and methods) and Figure 1 Graph E (Results) shows significant difference - The last sentence inform that all experiments are at least n=3.
Apologies for the confusion. In the human vs human MLR assay (Figure 1E) one healthy donor sample acts as the stimulator cell population and another healthy donor sample acts as the responder population. This is why two separate healthy donors are needed for each experiment. We have edited this method on line 441 to increase clarity.
Discussion: Please add your own opinion about results and the possibilities of using research results in clinical practice.
Authors' opinions have been added to the conclusion.
Conclusions: Consider rewording the conclusions to be clearly useful in further research on HCC pathogenesis or therapy, please.
We reworded the paragraph and added the future direction.
Reviewer 4 Report
This study reports -dependent effects of sorafenib on the immune response and attempts to liaise this to HCC development., 1 In vitro and in vivo experiments were performed with low and 20 high doses of sorafenib using human T cells and spontaneous developed woodchuck HCC models. Following exposure to the high dose of sorafenib the baseline activity of NFAT1 in T cells was significantly increased. A significant decrease in T cell proliferation and increased the proportion of PD-1-expressing CD8+ T cells with NFAT1 activation were detected in the presence of the high dose of sorafenib. In woodchuck, smaller tumors were detected in the low-dose sorafenib treated group compared to the placebo and high-dose treated groups. The low-dose sorafenib group showed a significant tumor growth delay compared to others with increased CD3+ cells in tumor. The authors propose that sorafenib had immunomodulatory effects with dose-and time-dependent manner. High, but not low, dose of sorafenib treatment was associated with an immunosuppressive action.
The study is overall sound and of significance for the field. The data support the conclusion. I have however several comments.
- there are a lot of inaccuracies in the text that need to be fixed. For instance Line 24"In THE in vivo model" etc; Line 52: "Sorafenib" etc. The authors have to proofread the manuscript thoroughly.
- Figure 2D: Tregs should be identified with CD4/CD25 but also FoxP3 positive staining. FoxP3 staining is missing.
- Figure 4 is totally black, it is not readable.
- there is a discrepancy between in vitro and in vivo data in that T cells that circulate are different of T cells that reside in the liver. Is not possible to do the same analyses done for human T cells, on blood T cells isolated from woodchuck?
Author Response
- there are a lot of inaccuracies in the text that need to be fixed. For instance Line 24"In THE in vivo model" etc; Line 52: "Sorafenib" etc. The authors have to proofread the manuscript thoroughly.
We are very sorry for multiple typos. Manuscript has now been proofread by multiple authors and corrections made accordingly.
- Figure 2D: Tregs should be identified with CD4/CD25 but also FoxP3 positive staining. FoxP3 staining is missing.
While we agree that FoxP3 can be a marker for Tregs, it has also been found to be expressed on non-Treg T cells.There are many studies supporting the phenotyping of T regs with CD25 alone. For this reason we believe that gating on CD4+, CD25 bright cells is sufficient to identify Tregs.
- Figure 4 is totally black, it is not readable.
We appreciate the quality issue with figure 4 that was echoed by other reviewer and we have now changed it as IHC.
- there is a discrepancy between in vitro and in vivo data in that T cells that circulate are different of T cells that reside in the liver. Is not possible to do the same analyses done for human T cells, on blood T cells isolated from woodchuck?
Unfortunately this is not possible yet. There is limited availability of commercial antibodies that cross-react with woodchuck antigens. In our studies only one such antibody could be found, a mouse anti-rat CD3 antibody that was utilized for immunofluorescence in tumor infiltration results. We have now added comments to this effect on lines 187 and 269. Our facility has analyzed woodchuck's DNA sequences, so we are hoping that we can generate new various antibodies for woodchucks soon.
Round 2
Reviewer 2 Report
The authors have better presented and commented on the results obtained and the conclusions of their work.
Author Response
Your comments were highly appreciated, and helped us to improve our manuscript.
Reviewer 4 Report
The authors did not fulfil one of my request. Their statement: "While we agree that FoxP3 can be a marker for Tregs, it has also been found to be expressed on non-Treg T cells.There are many studies supporting the phenotyping of T regs with CD25 alone. For this reason we believe that gating on CD4+, CD25 bright cells is sufficient to identify Tregs."
is not grounded. Please cite the pertinent literature stating the triple positive CD4+/CD25+/FoxP3 ARE NOT functional Tregs, or they are not the only ones., versus the opposite literature. Otherwise perform these experiments.
Author Response
We agree with the reviewer that a definitive identification of Tregs would require FoxP3 and (lack of expression of) CD127. However, the dilemma is that the ultimate functional phenotype of (Treg-associated) immunosuppression requires the isolation of viable cells that can be used in functional cell assays. The staining of FOXP3 requires cell fixation and permeabilization. Hence in the literature (including the provided reference) the application of CD4+/CD25br is used as criteria to isolate Tregs for functional assays. If CD4+/CD25br is used to isolate Tregs for functional immunosuppression assays and only following the functional assay are they identified as FoxP3+ it raises the question if the initial CD4+/CD25br in itself would be sufficient. Furthermore, there is literature that suggests heterogeneity of FoxP3 expression in the Tregs including FoxP3 negative cells (Devaud et al, 2014, see below). In order to accommodate the reviewer’s concern we edited the text in the Results and Discussion sections to emphasize that the CD4+/CD25br cells are consistent with Tregs and an immunosuppressive phenotype but that a definitive identification would require phenotyping for FoxP3.
The results (line 196-198) describe the CD4+/CD25br as being consistent with the Treg phenotype:
“A decrease in the proportion of CD4+CD25brightT cells, a phenotype consistent with regulatory T cells, was detected with sorafenib treatment in a dose-dependent manner (Figure 2D).”We now added a section to the discussion (lines 323-328 with track changes) to clarify that the CD4+/CD25br phenotype as applied in our study is associated with immunosuppressive function and that this is attributed to Treg cells and have included the relevant references : “ In the current study T cells were phenotyped for CD4 and CD25 co-expression; a phenotype of CD4 positive, CD25 bright is consistent with functional immunosuppression attributed to T regulatory cells (Mayer et al, 2012). The observed sorafenib-induced reduction of CD4+CD25brightTcells is in agreement with previously published literature [10]but is contradictory to others [41,42]. Definitive identification of this population as T regulatory cells would require additional functional markers, such as FoxP3 (Santegoets et al, 2015).
Devaud, C, Darcy, PK, Kershaw, MH. Foxp3 expression in T regulatory cells and other cell lineages. Cancer Immunol Immunother. 2014 Sep;63(9):869-76.